# Simulation of Photosynthetic Quantum Efficiency and Energy Distribution Analysis Reveals Differential Drought Response Strategies in Two (Drought-Resistant and -Susceptible) Sugarcane Cultivars

**DOI:** 10.3390/plants12051042

**Published:** 2023-02-24

**Authors:** Dongsheng An, Baoshan Zhao, Yang Liu, Zhijun Xu, Ran Kong, Chengming Yan, Junbo Su

**Affiliations:** 1South Subtropical Crop Research Institute, Chinese Academy of Tropical Agricultural Sciences, Zhanjiang Experimental and Observation Station for National Long-Term Agricultural Green Development, Zhanjiang 524091, China; 2Zhanjing Experimental Station, Chinese Academy of Tropical Agricultural Sciences, Zhanjiang 524013, China; 3Jiaxing Vocational and Technical College, Jiaxing 314036, China

**Keywords:** chlorophyll fluorescence, model, photoprotection, energy dissipation, water consumption

## Abstract

Selections of drought-tolerant cultivars and drought-stress diagnosis are important for sugarcane production under seasonal drought, which becomes a crucial factor causing sugarcane yield reduction. The main objective of this study was to investigate the differential drought-response strategies of drought-resistant (‘ROC22’) and -susceptible (‘ROC16’) sugarcane cultivars via photosynthetic quantum efficiency (*Φ*) simulation and analyze photosystem energy distribution. Five experiments were conducted to measure chlorophyll fluorescence parameters under different photothermal and natural drought conditions. The response model of *Φ* to photosynthetically active radiation (*PAR*), temperature (*T*), and the relative water content of the substrate (*rSWC*) was established for both cultivars. The results showed that the decreasing rate of *Φ* was higher at lower temperatures than at higher temperatures, with increasing *PAR* under well-watered conditions. The drought-stress indexes (*ε*_D_) of both cultivars increased after *rSWC* decreased to the critical values of 40% and 29% for ‘ROC22’ and ‘ROC16’, respectively, indicating that the photosystem of ‘ROC22’ reacted more quickly than that of ‘ROC16’ to water deficit. An earlier response and higher capability of nonphotochemical quenching (*NPQ*) accompanied the slower and slighter increments of the yield for other energy losses (*Φ*_NO_) for ‘ROC22’ (at day5, with a *rSWC* of 40%) compared with ‘ROC16’ (at day3, with a *rSWC* of 56%), indicating that a rapid decrease in water consumption and an increase in energy dissipation involved in delaying the photosystem injury could contribute to drought tolerance for sugarcane. In addition, the *rSWC* of ‘ROC16’ was lower than that of ‘ROC22’ throughout the drought treatment, suggesting that high water consumption might be adverse to drought tolerance of sugarcane. This model could be applied for drought-tolerance assessment or drought-stress diagnosis for sugarcane cultivars.

## 1. Introduction

Sugarcane is a major crop for sugar and bio-energy worldwide and essential for agricultural productivity and economic growth [1]. In China, sugar from sugarcane exceeds 90% of the total produced sugar; however, over 80% of the sugarcane planting area spans the slopes of arid lands in south China without irrigation facilities and where seasonal drought has become a major natural disaster. Seasonal drought is reflected in spatial–temporal characteristics based on the drought index of continuous days without precipitation [2]. Yield reduction caused by seasonal drought has gravely restricted the development of the sugarcane industry.

Several pot experiments and field studies have been conducted to understand the biological properties and improve the phenotypic and genotypic characteristics of sugarcane varieties under drought. Morphological parameters, including the leaf area, root and shoot weight, and stalk characteristics (length/height and diameter) of sugarcane, have been analyzed under drought induced via polyethylene glycol (PEG)treatment and withheld irrigation [3,4,5]. Both the drought-resistant and drought-susceptible genotypes showed decreased net photosynthetic rates (*P*_n_), stomatal conductance (*g*_s_), PSII quantum efficiency (*Φ*_PSII_), and leaf relative water content; however, the genotypes exhibited increased nonphotochemical quenching (*NPQ*), malondialdehyde (*MDA*), proline, superoxide dismutase (*SOD*),and ascorbate peroxidase (*APX*) contents, which were higher in drought-resistant than in drought-susceptible genotypes [6,7,8]. Yield component analysis confirmed that the drought-resistant variety exhibited higher productivity than did the drought-susceptible variety due to its superior stalk characteristics (height, weight, and number) [9]. Moreover, studies on the roles of specific candidate genes (belonging to the abscisic-acid (ABA)-dependent pathway) in the drought-stress responses of resistant and susceptible sugarcane clones laid the foundation and direction for improving drought resistance of sugarcanes through marker-assisted breeding [10].

Process-based models have also been developed to estimate photosynthesis, biomass production, dry matter partitioning, and sugar accumulation for informed decision-making in sugarcane production [11,12]. Combining the irrigation module and the gross margin calculator with the crop growth and water balance model could simulate the yield and survival of newly planted ratoon sugarcane under different water-allocation and climate scenarios, supported with field irrigation management [13]. The accuracy of biomass simulation was substantially increased via further improving the transpiration efficiency and root-length density aspects of the model. These factors are determined throughthe genetic variations of sugarcane to water deficit; hourly transpiration traits, such as midday flattening and the conductance limit; and the hydraulic conductivity in each soil layer [14]. Moreover, the sugarcane model could achieve remote and large-scale monitoring of sugarcane irrigation when improved with innovative techniques and methods. These techniques involve adding a thermal infrared index coupled with a remote sense of the fraction of intercepted photosynthetically active radiation (*fiPAR*), or satellite thermal infrared [15,16,17]. However, almost all models are inseparable from simulation of photosynthetic performance, which is essential to plant growth as the source of matter and energy.

As a photosynthesis probe, chlorophyll fluorescence (*ChlF*) parameters have been applied to measure photosystem operation efficiency rapidly and nondestructively under normal and stressful circumstances [18]. The photosystem is directly affected by drought via stomatal closure triggered by ABA signals transferred from the roots [19,20], and thus could be used to evaluate the performances of plants subjected to drought stress. Decreases in the maximum quantum efficiency of PSII photochemistry (*F*_v_/*F*_m_), *Φ*_PSII_, photochemical quenching (*q*_P_), and the electron transport rate (*ETR*) accompanied increases in nonphotochemical quenching (*NPQ*), fast-relaxing *NPQ* (*q*_E_), slowly relaxing *NPQ* (*q*_I_), and the yield induced via downregulatory processes (*Φ*_NPQ_); *Φ*_NO_ had been shown under drought stress in many studies [21,22,23,24].

Among the chlorophyll fluorescence parameters, the reliability of *F*_v_/*F*_m_ in evaluating the performance of sugarcane cultivars under drought stress was verified in a previous study [23]. The decreasing rate of *F*_v_/*F*_m_ was significantly higher in drought-tolerance cultivars than that in drought-susceptible cultivars under drought stress [25]. Though *F*_v_/*F*_m_ could become a stable drought-resistance indicator for sugarcane cultivars, considering its slow and slight decrease during drought stress (mostly 2~15%) [3,23,25], we chose to simulate another important parameter, *Φ* (namely *Φ*_PSII_), under various water statuses, and evaluated sugarcane drought tolerance through analysis of the fitted parameters for two reasons: (1) *Φ* provides information on the noncyclic electron transport rate through PSII [18] and photoinhibition based on inactivation of PSII reaction centers [26] and exhibits a large varied amplitude (47.5%~64.3% decrease) with aggravation of drought in C_4_ plants, including sugarcane and reed [24,27]; (2) *Φ* is directly related to the *ETR,* which provides energy to biochemical reactions of CO_2_ assimilation [28], thus having been used to estimate photosynthetic rates via the modified FvCB model, which describes the relationship between the rate of carboxylation for the C_4_ cycle (*V*_p_) and the rate of ATP production driven via e^-^ transport (*J*_ATP_), calculated from Δ*F*/*F*_m_′ (equal to *Φ*, where Δ*F* = *F*_m_′ − *F*_s_,) [29]. Two other parameters, *NPQ* and *Φ*_NO_, were chosen in this study to analyze energy distribution under drought. The former represented total energy dissipation into heat loss containing energy-dependent, zeaxanthin-dependent, and photoinhibitory quenching [30], and the latter has been used to evaluate the occurrence of physiological damage accumulation during drought [24] and post-drought recovery [31].

Thus, the main objective of this study was to establish the relationship between photosynthetic quantum efficiency and water status and reveal the different drought-response patterns in drought-tolerant and -susceptible cultivars via analyzing the energy distribution associated with drought stress. This study provides a new perspective for sugarcane drought-tolerance assessment and drought-stress diagnosis.

## 2. Results

### 2.1. Photothermal Model Parameterization and Its Biological Significance

*Φ* decreased with rising or falling temperature from *T*_o_ under three given *PAR* levels, and the decreasing rate intensified with increasing *PAR*. The *Φ*_bL_ values were lower than the *Φ*_bH_ values under all three given *PAR* levels, indicating that the sugarcane photosystem is more sensitive to low temperatures. All of the *Φ*-related parameters (*Φ*_bL_, *Φ*_bH_, and *Φ*_To_) declined with the increase in *PAR*, and the difference between *Φ*_bL_ and *Φ*_bH_ gradually increased, but both values occurred at the same *PAR* level in ‘ROC22’ and ‘ROC16’ (Figure 1). The *F*_v_*/F*_m_ value showed no significant difference between ‘ROC22’ (0.7693 ± 0.003) and ‘ROC16’ (0.7685 ± 0.0027). Parameter *Ε* reflected the amplitude decrease in *Φ*_To_ with increasing *PAR* and had the value of 5.91×10^−4^ for both tested cultivars (Table 1). In general, the cultivars exhibited high light efficiency at low *Ε*.

### 2.2. Response of the Estimated Parameter to Drought Treatment

The parameter *Ε*_D_ increased with the decrease in *rSWC* for both cultivars, and a higher *Ε*_D_ value of denoted severe effects of drought-induced stress on the plant thus could be used for drought-stress assessment. Different downward trends were observed between the two cultivars during drought stress. The *Ε*_D_ of ‘ROC22’ increased gradually when the *rSWC* decreased from 29% to 18%, while that of ‘ROC16’ increased drastically when the *rSWC* dropped below the 29% level. Based on the corresponding *Ε*
_D_ for drought-stress assessment, the slight, moderate, and severe drought conditions were described as 7 <*Ε*
_D_< 9, 9 <*Ε*
_D_< 13, and *Ε*
_D_> 13, respectively (Figure 2a).

According to our experiments, the *rSWC*_c_ values for ‘ROC22’ and ‘ROC16’ were 40% and 29%, respectively, showing that the photochemical reaction for ‘ROC22’ responded at a higher water status in contrast to that of ‘ROC16’ (Figure 2b). Additionally, *rSWC* decreased faster for ‘ROC16’ than for ‘ROC22’ during drought stress, indicating that the two cultivars possessed different water consumption characteristics under drought conditions.

### 2.3. Energy Distributionin PSII in Response to Drought

We used the parameters *Φ* and *NPQ* to assess the transfer of excitation light energy to photochemical reaction and its decay via heat loss, respectively. Five days after the drought treatment, the *Φ* of ‘ROC16’ was slightly higher than that of ‘ROC22’; however, both cultivars exhibited the same *NPQ* level on day 1, and it differed with the increasing *PAR* from day 3 to day 5 (Figure 3a–c). ‘ROC22’ maintained high *Φ* compared to ‘ROC16’ at day 7 after the drought treatment, but the two cultivars had the same *NPQ* level when the *PAR* was below 900 μmol photons m^−2^ s^−1^. After 9 days of drought stress, both the *Φ* and the *NPQ* of ‘ROC16’ were significantly lower than those of ‘ROC22’ (Figure 3d,e). The results demonstrated that the photoprotective mechanism based on the heat loss of ‘ROC16’ did not involve excessive energy consumptionwhen compared with that of ‘ROC22’. Another piece of evidence that showed less energy consumption was the increased *Φ*_NO_ levels observed on day 3 (*rSWC* 56%) and day 5 (*rSWC* 40%) for ‘ROC16’ and ‘ROC22’, respectively, after the drought treatment. The *Φ*_NO_ value of ‘ROC16’ was significantly higher than that of ‘ROC22’ after day 3, and the difference was magnified with aggravation of drought stress, indicating higher drought-induced damage imposed on ‘ROC16’ than on ‘ROC22’ (Figure 3f).

Furthermore, the *Ε*_D_ values of the two cultivars increased on day 5 after the drought treatment, but the *rSWC* values of ‘ROC22’ and ‘ROC16’ decreased to 40% and 29%, respectively. This differed from the *rSWC* values (56% for ‘ROC16’ and 40% for ‘ROC22’) that corresponded to the turning point of *Φ*_NO_, implying that the PSII of ‘ROC22’ responded to the water deficit more quickly than did that of ‘ROC16’. Contemporaneously, for ‘ROC22’, the *NPQ* rose to 1.6 when the *PAR* approached 1200 μmol photons m^−2^ s^−1^ but remained below 1.3 for ‘ROC16’ throughout the drought treatment. *NPQ* merely increased when the *PAR* ranged from 100 to 900 μmol photons m^−2^ s^−1^ on day 7 and day 9 for ‘ROC16’ and on day 5 for ‘ROC22’. This phenomenon demonstrated that rapid and high energy dissipation in drought is essential for drought tolerance in sugarcane.

### 2.4. Different Drought-Response Patterns for Two Cultivars

We investigated the diverse drought-response patterns of the two sugarcane cultivars. The results showed that ‘ROC22’ exhibited a quick response and a high capability of *NPQ* (Figure 2b) instead of liner electron transport (exhibited via low *Ε*
_D_) at the onset of drought (*rSWC* of 40%) (Figure 3c). This delayed the conversion from the drought-induced reaction to the physiological damage of the PSII, which was indicated with an increase in *Φ*_NO_ (Figure 3f). However, ‘ROC16’ maintained a relatively lower *Ε*
_D_ value to ensure photochemical efficiency until the *rSWC* dropped to 29%, after which the *NPQ* increased at *PAR* of merely 100~900 μmol photons m^−2^ s^−1^. Moreover, the *Φ*_NO_ of ‘ROC16’ was significantly higher than that of ‘ROC22’. The different water consumption characteristics of the two cultivars were revealed in the differentially decreasing rates of *rSWC*, which showed that ‘ROC16’ had higher *rSWC* than ‘ROC22’ throughout the drought treatment.

### 2.5. Model Validation

The coefficient of determination (*r*^2^) and relative root mean-squared error (*rRMSE*) of the predicted values were 0.922 and 0.11 (Figure 4a), while those of the measured values were 0.826 and 0.309 (Figure 4b), respectively, for *Φ* under different photothermal and drought conditions, indicating that the model could be applied for drought-tolerance assessment or drought-stress diagnosis for sugarcane cultivars.

## 3. Discussion

The minimum, optimal, and maximum temperatures obtained from the curve fitting of the net photosynthetic rate under saturated light (*P*_n,max_) and air temperature (*T*a) were used to describe the fundamental temperature for crop growth [32]. The relationship between *Φ* and *T* at limited *PAR* observed in our study was similar to those reported in previous studies [33]; however, in our study, the fitted parameters varied with different *PAR* levels (Figure 1). Unlike with *P*_n,max_, the occurrence of *Φ*_b_ (*Φ*_bL_ or *Φ*_bH_) at low or high temperatures was associated with alternative electron fluxes beyond photochemical reaction processes, such as photorespiration, the Mehler reaction, or cyclic and pseudocyclic electron transport under normal or drought conditions [34,35]. For instance, photorespiration and the Mehler reaction constituted 20% and 30% of the total electron flux, respectively [36,37]. Moreover, the *rETR* remained at 50% under normal conditions, even when photosynthesis ceased due to stomatal closure under drought stress [38]. This could also explain the existence of *Φ*_b_ and the gradual decrease in *Φ* with increases or decreases in temperature in our study compared with the *P*_n,max_-*T*a of the C_3_, C_4,_ and CAM plants [39].

The *Φ* of sugarcane, a typical C_4_ crop, showed higher sensitivity to lower temperatures than to higher temperatures because C_4_ pathway enzymes are cold-labile due to the limitations of phosphoenolpyruvate carboxylase (PEPC) and pyruvate phosphate dikinase (PPDK) [40,41]. This explains why the *Φ*_bL_ value was lower than that of *Φ*_bH_, and the increasing disparity between *Φ*_bL_ and *Φ*_bH_ with increasing *PAR*. The assumption that *Φ*_bL_ and *Φ*_bH_ exhibited the same downward trend as *Φ*_To_ in our study resulted in underestimation of *Φ*, especially when a value below 0.2 was obtained with a combination of drought, high *PAR,* and low (or high) temperature (Figure 4b). This phenomenon resulted from the fact that both net photosynthetic rate and *Φ*_PSII_ significantly decrease more under drought–cold stress than under drought stress only [42]. Additionally, combined heat and drought stress severely affect crop leaves at the physiological and biochemical levels via detrimental variation of photosynthetic pigments, osmolytes, and enzymatic antioxidant activities more than heat or drought stress alone [43,44].

We supposed that light energy was allocated between liner electron transport (*Φ*) and heat loss (*NPQ*) at the onset of drought stress until the occurrence of potential damage in the PSII, indicated with the concurrence of increased *Φ*_NO_ and decreased *NPQ* (or *q*_N_) shown in studies of salt stress [45,46] and disease influence [47]. The increase in *NPQ* compromised the decrease in *Φ* based on their new relationship acquired in the dark that followed illumination, which could be more appropriate to describe the photoprotective potential of *NPQ* [48]. This phenomenon also appeared in sweet sorghum and maize, with the degree of *NPQ* change differing in different cultivars under drought stress [49,50]. However, a significant increase in *Φ*_NO_ was found when sorghum suffered from drought [51]; this might be related to quenching of singlet oxygen via *β*-carotene [52], which could also cause photoinhibition via degradation of the D1 protein in the PSII center [53]. Similar results appeared in coastal halophytic marsh grasses with increasing salinity, and the high levels of the Mehler reaction and antioxidant enzymes improved adaptation in photoprotection [54]. Another study reported a decrease in *Φ*_NO_ with enhanced *Φ*_NPQ_ in maize under a long time field condition, which probably represented heat energy loss in protection of the stressed younger leaves [55].

Both the *Φ* and the water consumption of ‘ROC22’ were lower than those of ‘ROC16’ before the 5-day marker after drought, similarly to a previous study, where the sugarcane response to drought involved a rapid decrease in *P*_n_ and *g*_s_ for the drought-tolerance cultivar rather than the drought-sensitive one [7]. This might be associated with rapid stomatal closure, to reduce water loss, induced via ABA [56], which is synthesized in arid root systems to induce specific genes and proteins in response to water deficit [10,57]. In conversion of the value range of the *Ε*_D_ to water status, the drought degree of sugarcane plantation was divided into three stages: 29% < *rSWC* < 40% (slight), 18 < *rSWC*< 29% (moderate), and *rSWC* < 18% (severe). These ranges were lower than those reported in the previous study (which were 65~70% (slight), 45~50% (moderate), and 25~30% (severe)) [58], since we concentrated on the critical value of irrecoverable physiological injury instead of the yield reduction caused by the stress reaction. The accuracy, persuasiveness, and application scope of the model could be improved after more cultivars (or varieties) participate in parameter fitting and model validation.

## 4. Materials and Methods

### 4.1. Plant Materials and Treatment

Two sugarcane cultivars, ‘ROC22’ (drought-resistant) and ‘ROC16’ (drought-sensitive), cultivated by the Taiwan Sugar Research Institute, were used in this study; both were main cultivars introduced and applied in production in south China [59] and used in many drought-response researches [24,60,61]. Five experiments were conducted with the same planting, management, and drought treatment during different seasons in Zhanjiang, China (21° N, 110° E), from 2019 to 2021. Bucket planting was adopted for the experiment, with 30 buckets for each cultivar and three stems (1 bud per segment) per bucket, arranged based on field planting density (90,000 buds·ha^−1^). The caliber, bottom diameter, and height of each bucket were 40 cm, 30 cm, and 40 cm, respectively. The buckets contained red soil (70%), sand (20%), and organic manure (10%) as a substrate. The buckets were placed on hardened ground in an open environment without shading to implement photothermal experiments at the 8^th^/euphylla stage until the drought experiments were initiated. Five and ten buckets with uniform-growth seedlings were selected for the photothermal experiments and the drought experiments, respectively. The environmental conditions and substrate properties for each experiment are displayed in Table 2.

Photothermal experiments were conducted under field conditions. The average *F*_v_*/F*_m_ value of 9 leaves, measured at *T*_o_ under well-watered conditions and selected from 15 plants (5 buckets) in experiment 5 (*Exp5*) and with 30 points of *Φ* obtained under 3 given *PAR* levels (285, 625, and 1150 μmol photons m^−2^ s^−1^), with *T* ranging from 11.8 °C to 42.7 °C for each cultivar from 30 plants (10 buckets) in *Exp1-3*, was used to fit the photothermal response model (evaluation of *Φ* under different light and temperature backgrounds). Furthermore, 60 sets of the *Φ*, *PAR,* and *T* measured for each cultivar from*Exp4* and *Exp5*were used for model validation. The buckets were kept in a nearby greenhouse for a short time (10 min) for *Φ* acquisition at a higher *T* (over 38 °C).

For the subsequent drought experiments, the buckets were placed infield conditions, except for those in Exp5, which were moved into the phytotron from 6:00 to 12:00, with the PAR increasing from 0 to 1200 μmol photons m^−2^ s^−1^ (20% light intensity increased per hour using artificial light) under optimal *T* (29 ± 1.5 °C) at 1, 3, 5, 7, and 9 d after withholding of irrigation. The measured *Φ*, *VWC*, *F*_v_/*F*_m_, *PAR*, and *T* were used to estimate *Ε*_D_, and the energy dissipation parameters obtained from *Exp5* were used to analyze the different responses of the photoprotective mechanisms between the two cultivars under drought conditions. A total of 125 sets of the *Φ*, *VWC*, *F*_v_/*F*_m_, *PAR*, and *T* obtainedfor each cultivarfrom 8:00 to 12:00 am in *Exp1-4* were used for model validation.

### 4.2. Chlorophyll Fluorescence and Substrate Water Content Measurement

The chlorophyll fluorescence (*ChlF*) parameters were measured on the middle part of each +1 leaf (the first fully expanded leaf when counting from the top down) using a MINI-PAM-II analyzer (Walz, Effeltrich, Germany), which measures *PAR* and temperature via a self-contained sensor. All of the *ChlF* parameters were obtained under natural light conditions, except for those measured in the phytotron in *Exp5,* where artificial light had been used for light adaptation. The leaf clip was slightly adjusted to ensure relatively constant *PAR* (within ±10), and a stable *F*_s_ was waited for to initiate the process of measuring *F*_m_′. Then, *F*_o_ and *F*_m_ were measured after 30 min of dark adaptation using dark clips at the same leaf position. The next round of *F*_s_ and *F*_m_′ measurement was conducted after 30 min of natural light adaptation (artificial light in the phytotron in *Exp5*). Four basic fluorescence parameters, *F*_s_ (steady fluorescence from light-adapted leaves), *F*_m_′ (maximal fluorescence from light-adapted leaves), *F*_o_ (minimal fluorescence from 30 min dark-adapted leaves), and *F*_m_ (maximal fluorescence from 30 min dark-adapted leaves), were measured from morning to midday. The obtained parameters were then used to calculate the minimal fluorescence from light-adapted leaves (*F*_o_′) [62], photosynthetic quantum efficiency (*Φ*), the maximum quantum efficiency of PSII photochemistry (*F*_v_*/F*_m_), nonphotochemical quenching (*NPQ*), and the yield for other energy losses (*Φ*_NO_) [63] indicating non-regulated energy dissipation.

*F*_o_′ = *F*_o_/[(*F*_v_*/F*_m_) + (*F*_o_/*F*_m_′)]

*Φ* = (*F*_m_′ − *F*_s_)/*F*_m_′

*F*_v_*/F*_m_ = (*F*_m_ − *F*_o_)/*F*_m_

*NPQ* = *F*_m_/*F*_m_′ − 1

*Φ*_NO_ = 1/[*NPQ* + 1 + *q*_L_(*F*_m_/*F*_o_ − 1)], where *q*_L_ = (*F*_o_*′/F*_s_)(*F*_m_*′* − *F*_s_)*/*(*F*_m_*′* − *F*_o_*′*)

Parameter *Φ* was equivalent to and symbolized as Δ*F*/*F*_m_′, *Φ*_PSII_, *Φ*_II_, or *Φ*_2_ in different studies, with the same calculation result [21,22,23,24,27,29,31,45,46,47,48,49,51,54,55,63].

Volumetric water content (*VWC*) was measured at a depth of 0~16 cm in each bucket via AZS-100 (Aozuo, China) at 8:00 a.m., with 5 repetitions during the drought treatment. The *rSWC* was calculated with the formula *rSWC* = *VWC*/*VWC*_S_ (*VWC*_S_ was measured at sufficient irrigation with 3 repetitions; Table 1).

### 4.3. Model Construction

#### 4.3.1. Estimation of Photosynthetic Quantum Efficiency (*Φ*) under Different Photothermal Conditions

Three levels of *PAR* (285, 625, and 1150 μmol photons m^−2^ s^−1^) were adopted to measure *Φ* under different air temperatures (*T*) using artificial light in field conditions in different seasons, calculated using the formulae transformed from the calibration model proposed to evaluate the response of *Φ* to photothermal backgrounds [64] (Figure 1a):(1)$$\Phi (T)=\{\begin{array}{c} \Phi_{bL}+(\Phi_{To}-\Phi_{bL})\times \sin(\frac{\pi }{2}\times \frac{T-T_{\min}}{T_{o}-T_{\min}})\;T_{\min}\leq T\leq T_{o}\\ \Phi_{bH}+(\Phi_{To}-\Phi_{\mathrm{bH}})\times \sin(\frac{\pi }{2}\times \frac{T_{\max}-T}{T_{\max}-T_{o}})\;T_{o}\leq T\leq T_{\max} \end{array}$$
where *Φ*_*T*o_, *Φ*_bL_, and *Φ*_bH_ represent the optimum *Φ* under *T*_o_ and the basic *Φ* values under low *T* and high *T* at limited *PAR*. *T*_min_, *T*_o_, and *T*_max_ represent the minimal, optimal, and maximal temperatures for sugarcane photosynthesis and vegetative growth, with values of 15 °C, 30 °C, and 40 °C, respectively, according to previous studies [65,66,67]. The relationship between *Φ*_bL_, *Φ*_bH_, *Φ*_To_, and *PAR* was established as follows (Figure 1b):(2)$$\Phi_{\mathrm{bL}}=a_{L}\times e^{-PAR\times b_{L}}$$
(3)$$\Phi_{\mathrm{bH}}=a_{H}\times e^{-PAR\times b_{H}}$$
(4)$$\Phi_{To}=(F_{v}/F_{m})\times e^{-\varepsilon \times 10^{-4}\times PAR}$$

The mean value of the *F*_v_*/F*_m_ measured from *Exp1~3* was used for Equation (4)’s fitting, and *a*_L_, *b*_L_, *a*_H_, and *b*_H_ are empirical coefficients fitted from Equations (2) and (3). The parameter *Ε*, fitted from Equation (4), was merely related to different sugarcane cultivars.

#### 4.3.2. Estimation of the Parameters under Drought Conditions

We inserted Equations (2)–(4) into Equation (1) to determine the *Φ*, *F*_v_*/F*_m_, *PAR*, *T*, and *rSWC* measured under drought and used the values to describe the relationship between *ε* and *rSWC* (Figure 2a):(5)$$\varepsilon_{D}=\{\begin{array}{cc} \varepsilon & rSWC>rSWC_{c}\\ \varepsilon +c\times e^{-d\times rSWR} & rSWC\leq rSWC_{c} \end{array}$$
where *rSWC*_c_ indicates the critical limit of *rSWC* for the two sugarcane cultivars, *Ε*
_D_ is *Ε* under drought stress, and c and d are the empirical coefficients fitted from Equation (5).

The effect of drought stress on *Φ*_bL_ and *Φ*_bH_ was assumed to be the same as that of *Φ*_To_, and these values were calculated as:(6)$$\Phi_{\mathrm{bL}D}=\Phi_{\mathrm{bL}}\times e^{-\varepsilon \times 10^{-4}\times PAR}$$
(7)$$\Phi_{\mathrm{bHD}}=\Phi_{\mathrm{bH}}\times e^{-\varepsilon \times 10^{-4}\times PAR}$$
where *Φ*_bLD_ and *Φ*_bHD_ represent *Φ*_bL_ and *Φ*_bH_, respectively, under drought stress.

#### 4.3.3. Model Validation

The coefficient of determination (*r*^2^) and relative root mean-squared error (*rRMSE*) were adopted to analyze the conformity and accuracy between the predicted and the measured values and were calculated as follows:(8)$$r^{2}=\frac{{\left(\sum \left(x-\bar{x}\right)\left(y-\bar{y}\right)\right)}^{2}}{\sum {\left(x-\bar{x}\right)}^{2}\sum {\left(y-\bar{y}\right)}^{2}}$$
(9)$$rRMSE=\frac{\sqrt{\frac{\sum_{i=1}^{n}{\left(OBS_{i}-SIM_{i}\right)}^{2}}{n}}}{M}$$
where *x*, *y*, $\bar{x}$, $\bar{y}$, *OBS*_i_, *SIM*_i_, *M*, and *n* are the measured value, predicted value, average measured variables, average predicted variables, observed value, simulated value, average observed value, and sample size for Equations (8) and (9), respectively.

The definitions, units, and values (standard error shown in brackets) of the parameters and empirical coefficients are listed in Table 1.

## 5. Conclusions

The simulation and energy distribution analyses exhibited an opposition between water consumption and drought tolerance. ‘ROC22’ (drought-resistant) decreased its water consumption and activated a photoprotective mechanism to delay the drought-induced physiological injury, while ‘ROC16’ (drought-susceptible) maintained relatively high water consumption to guarantee photochemical reactions but was severely injured with aggravation of drought.

## Figures and Tables

**Figure 1 plants-12-01042-f001:**
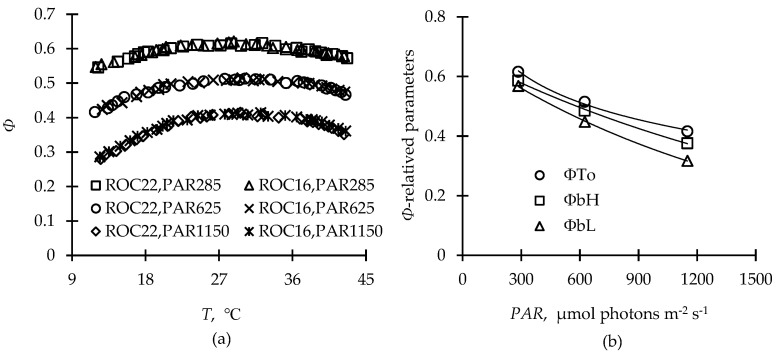
Response curves of photosynthetic quantum efficiency (*Φ*) to temperature (*T*) (**a**) and fitted curves of optimum *Φ* (*Φ*_To_) and basic quantum efficiency at low or high temperature (*Φ*_bL_ or *Φ*_bH_) to photosynthetically active radiation (*PAR*) (**b**).

**Figure 2 plants-12-01042-f002:**
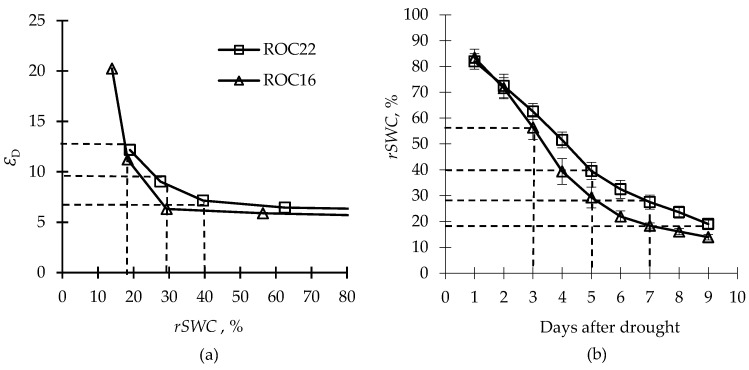
Response ofthe drought-stress index (*Ε*_D_) to the relative soil water content (*rSWC*), (**a**) and *rSWC* after drought treatment of the two sugarcane cultivars (**b**).

**Figure 3 plants-12-01042-f003:**
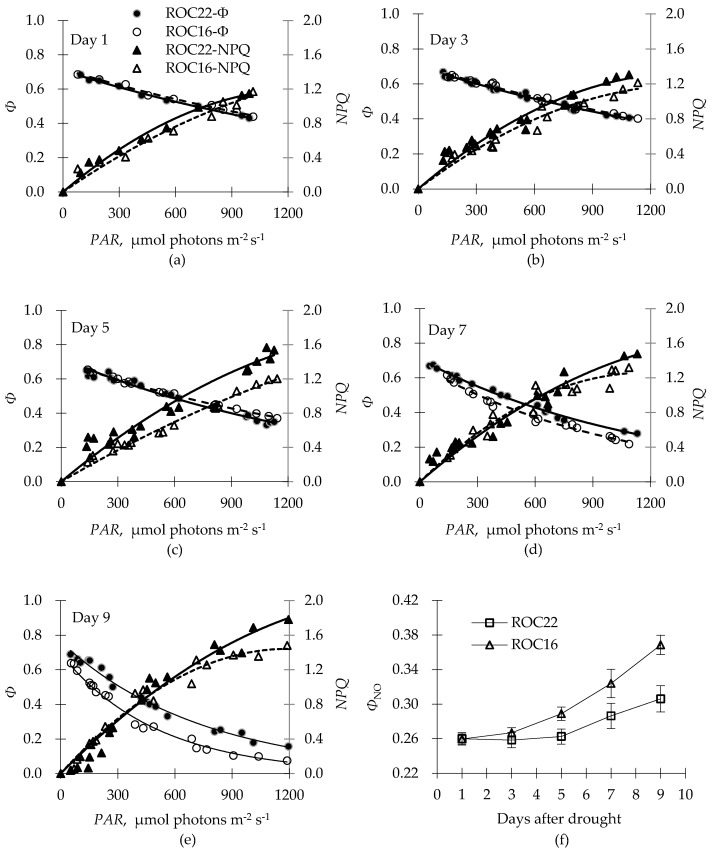
Excitation light-energy distribution in the two tested cultivars at 1, 3, 5, 7, and 9 days after drought treatment. (**a**–**e**) The photosynthetic quantum efficiency (*Φ*) and nonphotochemical quenching (*NPQ*) varied with *PAR* for two tested cultivars after withholding water ((**b**–**e**) share the same legend as (**a**)). (**f**) Average value of the yield for other energy losses (*Φ*_NO_) under different *PAR* vaired with the aggravation of drought stress for two tested cultivars.

**Figure 4 plants-12-01042-f004:**
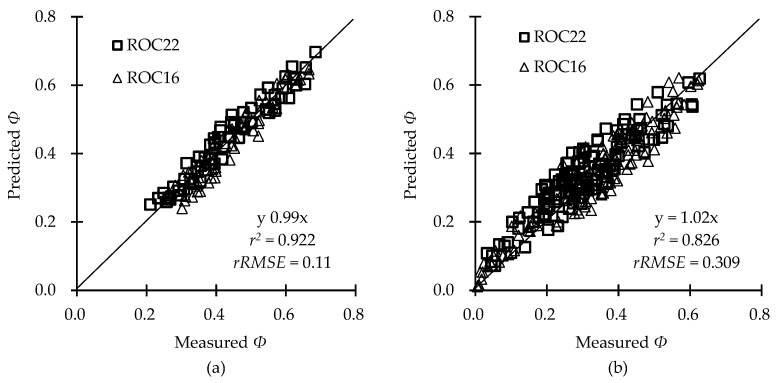
Comparisons between the measured photosynthetic quantum efficiency (*Φ*) and the predicted *Φ* of the two cultivars under different photothermal (**a**) and drought conditions (**b**).

**Table 1 plants-12-01042-t001:** List of parameters and empirical coefficients with definitions, units, and values (standard error shown in brackets) determined in this study.

Parameters	Definition	Unit	Value
*a*_L_, *b*_L_	Empirical coefficient describing the change of *Φ*_bL_ with *PAR*	—	0.688, 6.77 × 10^−4^ for both cultivars
*a*_H_, *b*_H_	Empirical coefficient describing the change of *Φ*_bH_ with *PAR*	—	0.676, 5.16 × 10^−4^ for both cultivars
*c*, *d*	Empirical coefficient describing the change of *Ε*_D_ with *rSWC* when *rSWC* was below *rSWC*_c_	—	39.63, 10.01 for ROC22 286.78, 21.53 for ROC16
*F*_v_/*F*_m_	Maximum quantum efficiency of PSII photochemistry	—	0.769 for both cultivars without water deficit
*NPQ*	Nonphotochemical quenching	—	—
*PAR*	Photosynthetically active radiation	μmol photons m^−2^ s^−1^	—
*rSWC*	Relative substrate water content calculated from *VWC*/*VWC*_s_	%	—
*rSWC* _c_	Critical value of *rSWC*	%	40 for ROC22 29 for ROC16
*T*	Temperature	°C	—
*T* _min_	Minimum temperature for sugarcane photosynthesis and growth	°C	15
*T* _o_	Optimum temperature for sugarcane photosynthesis and growth	°C	30
*T* _max_	Maximum temperature for sugarcane photosynthesis and growth	°C	40
*VWC*	Volumetric water content of substrate	%	—
*VWC* _s_	Saturated *VWC*	%	38 (0.66) for *Exp1~3* 38.3 (1.11) for *Exp4~5*
*Ε*	The amplitude decrease in *Φ*_To_ with increasing *PAR*	—	5.91 × 10^−4^ for both cultivars
*Ε* _D_	*Ε* varied with *rSWC* to describe the drought-stress degree	—	—
*Φ*	Photosynthetic quantum efficiency	mol e^−1^ (mol photon)^−1^	—
*Φ* _bL_	Basic *Φ* under low temperature, varied with *PAR*	mol e^−1^ (mol photon)^−1^	—
*Φ* _bH_	Basic *Φ* under high temperature, varied with *PAR*	mol e^−1^ (mol photon)^−1^	—
*Φ* _To_	Optimum *Φ* under *T*_o_, varied with *PAR*	mol e^−1^ (mol photon)^−1^	—
*Φ* _NO_	Yield for other energy losses (indicate non-regulated energy dissipation)	mol e^−1^ (mol photon)^−1^	—

**Table 2 plants-12-01042-t002:** Detailed information on the experimental conditions and physicochemical properties of the substrate.

		*Exp*1 ^aM, bV^	*Exp2* ^aM, bV^	*Exp3* ^aM, bV^	*Exp4* ^aV, bV^	*Exp5* ^aV, bM^
Experimental and Environmental Conditions						
Photothermal Experiment	Start date	9 April 2020	11 July 2020	30 November 2020	20 May 2021	23 October 2021
	Finish date	11 April 2020	13 July 2020	2 December 2020	22 May 2021	25 October 2021
	Average daily high temp (°C)	24.3 ± 2	36.2 ± 1.3	21.7 ± 1.8	35.2 ± 1.7	23.3 ± 4
	Average daily low temp (°C)	19.4 ± 1.4	28.6 ± 0.2	11.4 ± 3.3	27.5 ± 0.1	16.7 ± 1.1
	Average daily *PAR*_ave_ (μmol photons m^−2^ s^−1^)	233 ± 129	575 ± 61	364 ± 27	523 ± 139	314.6 ± 165
	Average daily *PAR*_max_ (μmol photons m^−2^ s^−1^)	1477 ± 623	2119 ± 66	1573 ± 231	2085 ± 253	1474 ± 647
Drought Experiment	Start date	13 April 2020	14 July 2020	3 December 2020	23 May 2021	26 October 2021
	Finish date	22 April 2020	20 July 2020	14 December 2020	2 June 2021	4 November 2021
	Average daily high temp (°C)	26.6 ± 2.5	35.3 ± 1.2	23 ± 1.3	34.8 ± 1.2	28.2 ± 0.8
	Average daily low temp (°C)	20 ± 4.7	27.1 ± 1	15.3 ± 2.8	26.9 ± 1.1	21.4 ± 2.1
	Average daily *PAR*_ave_ (μmol photons m^−2^·s^−1^)	341 ± 124	535 ± 60	262 ± 93	498 ± 74	321 ± 81
	Average daily *PAR*_max_ (μmol photons m^−2^ s^−1^)	1896 ± 388	2313 ± 138	1441 ± 322	2412 ± 208	2012 ± 311
Physicochemical Properties of the Substrate						
	Bulk density (g·cm^−3^)	1.19	1.19	1.19	1.21	1.21
	Saturated volumetric water content (%)	38	38	38	38.3	38.3
	Organic C (g·kg^−1^)	17.51	17.51	17.51	19.22	19.22
	Avaliable N (mg·kg^−1^)	36.34	36.34	36.34	33.88	33.88
	Avaliable P (mg·kg^−1^)	15.62	15.62	15.62	12.96	12.96
	Avaliable K (mg·kg^−1^)	44.70	44.70	44.70	46.16	46.16
	pH	4.82	4.82	4.82	5.07	5.07

Note: ^a^ and ^b^ denote photothermal and drought experiments, while ^M^ and ^V^ denote experimental data used for modeling and validation, respectively. Temperature and *PAR* data were selected from the daily data set recorded with a farm land environment monitoring system (CR1000).

## Data Availability

The data presented in this study are available upon request from the corresponding authors.

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
