# Peer review of "Simulation of Photosynthetic Quantum Efficiency and Energy Distribution Analysis Reveals Differential Drought Response Strategies in Two (Drought-Resistant and -Susceptible) Sugarcane Cultivars"

_plants, 2023, doi:10.3390/plants12051042_

Round 1
Reviewer 1 Report
First of all, I would like to state that the method of the experiments presented in the manuscript and the resulted model are interesting and novel in several aspects. Such models can be used to improve crop technology, to further develop the model to make predictions (e.g. on the effects of different degrees of drought stress), to integrate it into decision support systems for agriculture (e.g. impulses for irrigation regulation).
However, the manuscript is not suitable for publication in its present form, as it requires major revisions, with inevitable additions.
In addition to a number of minor to moderate errors (examples detailed below), the OVERALL PROBLEMS with the manuscript are:
1. The structure of the information (chapters) is not clear, The Results include information for the Materials and Methods section, and the Discussion section mostly re-states the own results, with a small amount of relevant literature comparisons. Similarly, a Conclusions chapter is also very poor.
Subchapters 2.5.1-3 are given in the results section, but methodological information is included there, and the equations are not referenced. (Or are they all their own equations???)
2. The method of the experiment is very incompletely described and thus not replicable (missing meteorological data series and parameters of the root medium relevant for the experiment, microclimate data at the time of the measurements, exact method of taking the measurements; we do not know how many plants were actually measured. How were the three basic PAR values and, for example, the minimum temperature selected?)
3. The authors have used a lot of parameters in the study, calculations have been made (some possibly by the fluorometer itself), but in most cases the meaning of the subscript of the name of the parameter is not clear, making interpretation difficult (in many cases impossible to understand). In all cases, the parameters (including all their variants indexed) must be precisely defined in the substance and method section.
4. While the introduction chapter is nicely written, there is a lot of specific information missing, which presumably the authors used in designing the experiment. For example, a summary of the environmental requirements of sugar cane (mainly related to temperature, light, water supply) and the cultivar types (drought sensitivity), supported by references to the literature, is missing. Also, there is no source indicated for the drought tolerance of the two varieties studied, although they are reported in several scientific publications.
I missed, for example, the following publication from the literature list: Khalil, F.; Naiyan, X.; Tayyab, M.; Pinghua, C. Screening of EMS-Induced Drought-Tolerant Sugarcane Mutants Employing Physiological, Molecular and Enzymatic Approaches. Agronomy 2018, 8, 226. https://doi.org/10.3390/agronomy8100226.
There are also no references to the formulae used for model calculation (subsections 2.5.1-3).
In my opinion, the model can be set up by knowing the behaviour of several species (for example 3 drought resistant and 3 -susceptible). If it is not possible to set up the model in an optimum size experiment, several cultivars (at least 3 resistant and 3 susceptible varieties) should be used in the validation phase, the back-testing phase, before the model is published. This would probably require the authors to carry out a further validation experiment.
All this leads me to believe that to make major changes to the manuscript that are larger than a "major" revision, in addition to the major changes to the manuscript, additional results need to be added (another validation experiment). In my opinion, only then will the material be worth publishing in the ‘plants’ journal.
If the editor and the other opponents consider that the manuscript can be published after revision, I suggest that, in addition to the above, the following comments be considered:
Title: It is too long and suggests (due to the specificity of the English language) that several cultivars have been studied, while the authors have constructed a model based on data from 1 - 1 cultivars.
Abstract: very long, more than twice the required length (the journal requirement is "about 200 words maximum"). The structure is very complicated, it is not clear what is the method of the experiment and what are the results (temperature values), and rather summarized results are listed.
Keywords: all keywords entered are included in the title. It is recommended to include keywords that are not in the title or abstract.
Are the temperature values mentioned in lines 23-24 (abstract) and lines 129-131 (results) the results of the experiment or a part of the method adopted from others? (no reference). If a result, then: e.g. how do we know that the lowest value is really a temperature minimum based on photosynthetic quantum efficiency? It seems, they were not measurements below this temperature (what could show maybe more low efficiency).
In the Table 2 (which is in the text wrongly cited like Table 1), it is not clear, what does it mean „Hight temperature” and „Low temperature”. In the photothermal measurements (and also in the other part) which were the environmental conditions, which is imported for the plant reactions (adaptation), which were the settings of the MINI-PAM II chamber during the measuring (chamber and/or environmental temperature during the dark adaptation). If more measuring (more temperature or light condition in the chamber) was established in one day, between two measuring was adaptation time or break for the physical processes to return to the „normal” level. (Unfortunately, the method description is so poor, so not clear, that I only try deduce that was the experiment.)
In case of the thermal effects and drought-stress condition measurements, we should know more about the root substrate heat and water household properties (in this situation there are more important, than the C, N,P, K content or pH). In this Table, also is not clear the „note” (legend).
My understanding is that there were not two experiments each time (a photothermal and a drought-stress), but that the researchers measured some parameters/response of the plants before inducing drought stress. (If I understand correctly, the same plant populations were tested.) If this was really the case, then Table 2 should not give the impression that there were two experiments each time.
It is not clear whether measurements were also taken under optimal water supply (as a control), and if so, what were the results?
Relative soil water content – is not a correct term, because the plants were grown in an special medium. How was the parameter measured/calculated and what exactly does the depth 0-16 cm mean (row 333)?
Row 324 – what temperature were the plants kept at (constant 29°C, all day?), how intense was the light?
Row 145: based on the Fig 2a, the rSWC in case of the ROC16 was not decreased under 18%! (So no conclusions can be told from this range.)
Discussion: in its present form, not specific enough (references are not closely linked enough, they are general). Additional, more relevant sources are needed, and the own results should not be reported or repeated in this chapter.
The figures and tables are not in the correct place and order (need be immediately after the first mention in the paragraph) in the manuscript. For example, the order of reference in the text for the figures is: 1, 4a, 2a, 2b…
Table 1: the title is wrong (it's not a list of parameters, but mainly values), and the explanation of the symbols is missing
Table 2: legend is not clear
Figure 1a: half of the legend is not visible
Figure 2b: x-axis: days after treatment – what was the treatment??? last irrigation?
Figure 3: presented together the first 4 experiment? The PAR values were the current PAR of natural light? or sets in the instrument / measure chamber?
In the version I have read, there are black and white figures with indistinguishable data series labels.
References: it is necessary to unify the format, as required by the journal.
Author Response
Dear editor and reviewer,
Thanks for the editor’ and reviewers’ opinions, which are very helpful to improve the quality of the manuscript and the growth of young scientific personnel. We have carefully restructed the article, add explaination of the cultivars choice and experiment conduction, and a description table for all the parameters used in this study. Words in blue a the changes I have revised in the manuscript. Now I response the reviewers’ comments with a point by point and high light the changes in revised manuscript. We sincerely hope that you find our responses and modificatins satisifactory are acceptable for publication. Full details are listed below:
Point 1: The structure of the information (chapters) is not clear, The Results include information for the Materials and Methods section, and the Discussion section mostly re-states the own results, with a small amount of relevant literature comparisons. Similarly, a Conclusions chapter is also very poor. Subchapters 2.5.1-3 are given in the results section, but methodological information is included there, and the equations are not referenced. (Or are they all their own equations???)
Response 1: We restructed the entire article to make each part clearly, especially among the Results, Materials and Method, Discussion sections. I add the relveant literature comparisons between our results and other studies in changes of chlorophyll fluorescence parameters, then concise the conclusions. I have described the constructed models in Materials and Method section instead of Result section, the transforming equations describing the relationship between photosynthetic quantum efficiency (Φ) and photo-thermal background were based on my dissertation which has been cited in the revised manuscript. The relationship between estimated parameters and water status was newly proposed in this study.
Point 2: The method of the experiment is very incompletely described and thus not replicable (missing meteorological data series and parameters of the root medium relevant for the experiment, microclimate data at the time of the measurements, exact method of taking the measurements; we do not know how many plants were actually measured. How were the three basic PAR values and, for example, the minimum temperature selected?)
Response 2: We rewrite the Method section, devided the meteorological data into two parts, for photo-thermal experiments and droght experiments, respectively. The environmental data acquired from CR1000 at meteorological observation field, while the microclimate data at the time of the measuremets obtained by the sensors of the fluorometer. 5 buckets (15 plants) and 10 buckets (30 plants) with uniform growth seedlings had been selected to conduct the photo-thermal experiments and droght experiments. Indeed there is dispute of basic values of sugarcane photo-thermal limination, we intended to use our data at first, after reading the expert’s queries, we chose to summarize the basic three point temperature of sugarcane from several studies which determined as 15℃(minimum), 30℃(optimum), 40℃(maximum). Furthermore, curve fitted parameters ΦTo, ΦbL, and ΦbH, which represents the optimum Φ under To, the basic Φ under low T and high T at the limited PAR based on the new three point temperature and recaculated all the data.
Point 3: The authors have used a lot of parameters in the study, calculations have been made (some possibly by the fluorometer itself), but in most cases the meaning of the subscript of the name of the parameter is not clear, making interpretation difficult (in many cases impossible to understand). In all cases, the parameters (including all their variants indexed) must be precisely defined in the substance and method section.
Response 3: I have filled all the parameters used in this study into a table containing the name, definition, unit and value to make them clearly understand.
Point 4: While the introduction chapter is nicely written, there is a lot of specific information missing, which presumably the authors used in designing the experiment. For example, a summary of the environmental requirements of sugar cane (mainly related to temperature, light, water supply) and the cultivar types (drought sensitivity), supported by references to the literature, is missing. Also, there is no source indicated for the drought tolerance of the two varieties studied, although they are reported in several scientific publications.
I missed, for example, the following publication from the literature list: Khalil, F.; Naiyan, X.; Tayyab, M.; Pinghua, C. Screening of EMS-Induced Drought-Tolerant Sugarcane Mutants Employing Physiological, Molecular and Enzymatic Approaches. Agronomy 2018, 8, 226.https://doi.org/10.3390/agronomy 8100226
There are also no references to the formulae used for model calculation (subsections 2.5.1-3)
Response 4: I delet the agronomic and engenieering water saving discription instead of adding many references of the chorophyll fluorescence parameters performance and their mechanism under drought to enrich the Introduction and to explain how we chose prarmeters for simulation. The logical consecution described as: (1) sugarcane importance, (2) problem (drought) in industry, (3) biological properties under drought, (4) application of sugarcane growth and irrigarion model, (5) importance of photosynthetic performance simulation in model, (6) chlorophyll fluorescence research revealed photosynthetic performance, (7) choice of parameter and objective of our sudy.
I supplimant the references of the properties of tested cultivars in Material section to explain why we chose these two cultivars.
The transforming equations describing the relationship between photosynthetic quantum efficiency (Φ) and photo-thermal background were based on my dissertation which has been cited in the revised manuscript. The relationship between estimated parameters and water status was newly proposed in this study.
Point 5: In my opinion, the model can be set up by knowing the behaviour of several species (for example 3 drought resistant and 3 -susceptible). If it is not possible to set up the model in an optimum size experiment, several cultivars (at least 3 resistant and 3 susceptible varieties) should be used in the validation phase, the back-testing phase, before the model is published. This would probably require the authors to carry out a further validation experiment.
Response 5: Indeed, the more cultivars used for validation, the more robust the model is. However, we chose these two tested cultivars considering the following points: (1) they were introduced and applied in production in south China as main cultivars (Wei, 2021); (2) they were widely used in many drought response researches (Yao, 2013; An,2015; Khalil, 2018). Through the literature review, 3, 1, 2, 1 of 7 references use 2, 3, 4, 6 cultivars for experiment. Besides, the more cultivars used in experiment, the less treatment levels conducted in experiment. Amount of time for light and dark adapetd was needed in our study due to the demand of chlorophyll fluorescene parameters measurement. We believe that two main cultivars with significant difference in drought tolerance could be representative and able to achieve our aim.
These are the referances:
- IACSP94-2094 (resistant) and IACSP97-7065 (sensitive) had been used in South American for drought and chilling study. (Sales, 2013)
- IAC91-2195 (sensitive) and IAC91-5155 (tolerant) had been used in Brazil to measure leaf growth rate under four water status (Queiroz, 2011).
- KK3 (tolerant) and K93-219 had been used in Thailand to measure plant growth, physiological characters and chlorophyll fluorescence parameters (Bamrungrai, 2021).
- SP83-2847, SP83-5073 (tolerant) and SP90-3414, SP90-1638 (sensitive) had been used in Brazil to measured the physiological indices and antioxidant enzymes at 3, 10, 20 days afger withholding water (Cia, 2012).
- KK3 (tolerant), LK92-11 (moderate) and K88-92 (moderate) had been used in Thailand to measure the plant growth, dry matter and Fv/Fm (Khonghintaisong, 2017).
- RB72454 and RB855453 (prone), SP81-3250 and SP83-2847 (tolerant) had been conducted in Brazil with two water regime and twice measurements (28 day and 56 day) of chlorophyll content, dry matter and Fv/Fm (Silva, 2018).
- Six cultivars have been used for agronomic characters, plant dry matter, and the maximum photosynthesis rate in at drought and control conditions (Chanokat, 2020).
Point 6: Title: It is too long and suggests (due to the specificity of the English language) that several cultivars have been studied, while the authors have constructed a model based on data from 1 - 1 cultivars.
Response 6: We revised the title to make it clear.
Point 7: Abstract: very long, more than twice the required length (the journal requirement is "about 200 words maximum"). The structure is very complicated, it is not clear what is the method of the experiment and what are the results (temperature values), and rather summarized results are listed.
Response 7: I refined the abstuact to narrate the methods and main conclusions. Three basic temperature reacquird from the summary of referances and used to fit the Φ-related parameters.
Point 8: Keywords: all keywords entered are included in the title. It is recommended to include keywords that are not in the title or abstract.
Response 8: The keywords has been revised by using the imoprtant conceptions not included in the title.
Point 9: Are the temperature values mentioned in lines 23-24 (abstract) and lines 129-131 (results) the results of the experiment or a part of the method adopted from others? (no reference). If a result, then: e.g. how do we know that the lowest value is really a temperature minimum based on photosynthetic quantum efficiency? It seems, they were not measurements below this temperature (what could show maybe more low efficiency).
Response 9: The experts’ opinion is very robust, indeed we did not known how the plants performance if temperature is lower or higher. It is inappropriate to determine the basic temperature through our experiment. Thus, we chose to summarize the basic three point temperature of sugarcane from several studies which determined as 15℃(minimum), 30℃(optimum), 40℃(maximum). Furthermore, curve fitted parameters ΦTo, ΦbL, and ΦbH, which represents the optimum Φ under To, the basic Φ under low T and high T at the limited PAR based on the new three point temperature and recaculated all the data.
Point 10: In the Table 2 (which is in the text wrongly cited like Table 1), it is not clear, what does it mean „Hight temperature” and „Low temperature”. In the photothermal measurements (and also in the other part) which were the environmental conditions, which is imported for the plant reactions (adaptation), which were the settings of the MINI-PAM II chamber during the measuring (chamber and/or environmental temperature during the dark adaptation). If more measuring (more temperature or light condition in the chamber) was established in one day, between two measuring was adaptation time or break for the physical processes to return to the „normal” level. (Unfortunately, the method description is so poor, so not clear, that I only try deduce that was the experiment.)
Response 10: The environmental conditions in Table 1 discribed the average daily value of PAR and temperature during each experiment. All experiment conducted in field conditon growth under natural environment, except Exp5, which also grew under field condition but moved to control environment at 1,3,5,7,9 day after withhold water only to acquire the ChLF data under optimum temperature used for curve fitting. From the morning to midday, The Fs and Fm’ were measured under limited PAR after Fs is stable after 30 min light adaption, then at same leaf position, Fo and Fm were measured after 30 min dark adaption had been adopted by using the dark clips.
Point 11: In case of the thermal effects and drought-stress condition measurements, we should know more about the root substrate heat and water household properties (in this situation there are more important, than the C, N,P, K content or pH). In this Table, also is not clear the „note” (legend).
Response 11: We have measured water household properties, which was now displayed in the table 2, but lack of heat. Two values of VWCs respresented twice substrate mixing, one for Exp1~3, another for Exp4~5.
Point 12: My understanding is that there were not two experiments each time (a photothermal and a drought-stress), but that the researchers measured some parameters/response of the plants before inducing drought stress. (If I understand correctly, the same plant populations were tested.) If this was really the case, then Table 2 should not give the impression that there were two experiments each time.
Response 12: The photo-thermal experiment and drought stress experiment were continuous, the latter was after the former. 5 buckets and 10 bucket with uniform growth seedling selected from the planted 30 buckets were used for photo-thermal experiment and drought stress experiment respectively. After acquired the measurement under different photo-thermal conditions, followed the drought treatment for each set of experiment. There are 5 set of experiments.
Point 13: It is not clear whether measurements were also taken under optimal water supply (as a control), and if so, what were the results?
Response 13: The measurements from day 1 after withholding water can be deemed as optimal water supply since the rSWC of which were over 80% for both cultivars, according to previous studies (Queiroz, 2011; Cia, 2012; Jin, 2012).
Point 14: Relative soil water content – is not a correct term, because the plants were grown in an special medium. How was the parameter measured/calculated and what exactly does the depth 0-16 cm mean (row 333)?
Response 14: We made a mistake of the incorrect discription at here and modify this term to relative water content of substrate with the same abbreviation rSWC. The depth 0~16cm means the VWC we measured is the average value of 0~16cm substrate water status resulted from the prob length of our equipment is 16cm, which inserted in substrate vertically.
Point 15: Row 324 – what temperature were the plants kept at (constant 29°C, all day?), how intense was the light?
Response 15: This setting was only for the standred experiment in Exp5 under controlled envrionment to obtained the model construction data. We add the specific settings in Materials and Method section: the buckets were moved into the phytotron from 6:00 to 12:00 with the PAR increasing from 0 to 1200 μmol·m-2·s-1 (20% light intensity increased per h) under optimal T (29±1.5 ℃).
Point 16: Row 145: based on the Fig 2a, the rSWC in case of the ROC16 was not decreased under 18%! (So no conclusions can be told from this range.) 22
Response 16: There is no data measured after rSWC decreased below 18%, thus I fully accepted the experts’ comment and delete this sentence to make the result rigorous.
Point 17: Discussion: in its present form, not specific enough (references are not closely linked enough, they are general). Additional, more relevant sources are needed, and the own results should not be reported or repeated in this chapter.
Response 17: I rewrite the Discussion section, add many references to discuss the variation of ChlF parameters under drought comparing with other researches, and move the cross analysis to Result 2.4.
Point 18: The figures and tables are not in the correct place and order (need be immediately after the first mention in the paragraph) in the manuscript. For example, the order of reference in the text for the figures is: 1, 4a, 2a, 2b…
Response 18: Accepted the experts’ comment and journal requirement to revise the manuscript.
Point 19: Table 1: the title is wrong (it's not a list of parameters, but mainly values), and the explanation of the symbols is missing,
Response 19: I have revised the table with all the parameters used in this study containing the name, definition, unit and value to make them clearly understand.
Point 20: Table 2: legend is not clear
Response 20: Accepted the experts’ comment and revised carefully
Point 21: Figure 2b: x-axis: days after treatment – what was the treatment??? last irrigation?
Response 21: have been revised to “days after drought”
Point 22: Figure 3: presented together the first 4 experiment? The PAR values were the current PAR of natural light? or sets in the instrument / measure chamber?
Response 22: The data used for energy dissipation analysis came from Exp5 at control conditions. The PAR value contained both light from the phytotron light source and instrument source, bucause the light intensity in pytotron varied with the distance between the leaf and light source unlike the sunlight. The light from the instrument was needed as a suppliment and all the measurements of Φ were at the stable Fs under the almost constant PAR.
Point 23: In the version I have read, there are black and white figures with indistinguishable data series labels.
Response 23: The Φ values measured at same given PAR showed no significant difference between two cultivars, which made many point overlap in the data series.
Point 24: References: it is necessary to unify the format, as required by the journal
Response 24: Accepted the experts’ comment and revised carefully

Reviewer 2 Report
Detailed comments:
Introduction: more introduction is necessary about the photosynthesis and the drought stress. Drought stress impact the whole photosynthetic apparatus and photosynthetic electron transport chain, not only the photosystems
The aims are too general and do not address specific novelty of the study.
Parameters PhibL, PhibH etc. - these parameter should be clearly defined at its first mention.
line 131 'The Fv/Fm value and empirical coefficients...' these parameters should be described separately for better understanding.
line 133 'Parameter Epsilon...' this has to be defined.
line 162-164 'Furthermore, the fitted NPQ curve for ‘ROC16’ exhibited a flattening trend with the increasing PAR, demonstrating that the photo-protective mechanism did not involve excessive energy consumption.' This sentence is unclear.
line 165: The PhiNO is not defined. To interpret these results, the concept of regulated and non-regulated non-photochemical quenching must be introduced.
why is a separate section 2.4 necessary? The figures should be placed where the relevant results are mentioned in the Results section.
Figure 1a - there are undefined symbols in the figure (what do the x, diamonds etc represent).
Table 1 - why is Fv/Fm given in 4 decimals? This is uncommon and misleading.
2.5 Model description. It is unclear why is this model selected and what was the rationale and background of its application. Therefore more information about the model and the reason of its application should be added.
line 234: Minimum, optimal, and maximum temperatures obtained from the curve fitting... how was this done?
line 238-240: it is unclear what was the rationale and background of dividing quantum yield to 'basic' and 'variable' quantum yield. Furthermore, it is unclear why would basic quantum yield represent alternative electron flow. Alternative electron flow activity should be determined with simultaneous measurements of Chl fluorescence and oxygen uptake. It should be explained how the alternative electron flow pathways were determined using other combined techniques e.g. oxygen polarography and Photosystem I kinetics, because these pathways require the application of these combined methods and integrated approaches. Therefore the interpretation of section 243-248 should be reconsidered and rewritten accordingly.
line 305-306: why these cultivars? What is the novelty and significance of plant material selection?
line 314-315: it is unclear how were the drought experiments done.
line 319: photo-thermal response model - this should be defined and described in details.
line 326: drought response model - more details are needed about this model.
line 326-327: 'The energy dissipation parameters were used to analyze the photo-protective capacity of the model' describe this in details
line 339-347: It is somewhat unclear what was the rationale of the authors for using the selected chl fluorescence parameters. This could be clearer if there was a proper introduction about photosynthesis and drought stress.
Author Response
Dear editor,
Thanks for the editor’ and reviewers’ opinions, which are very helpful to improve the quality of the manuscript and the growth of young scientific personnel. We have carefully restructed the article, add explaination of the cultivars choice and experiment conduction, and a description table for all the parameters used in this study. Words in blue a the changes I have revised in the manuscript. Now I response the reviewers’ comments with a point by point and high light the changes in revised manuscript. We sincerely hope that you find our responses and modificatins satisifactory are acceptable for publication. Full details are listed below:
Point 1: Introduction: more introduction is necessary about the photosynthesis and the drought stress. Drought stress impact the whole photosynthetic apparatus and photosynthetic electron transport chain, not only the photosystems
Response 1: We accepted expert’ opinion to add the references of the chorophyll fluorescence parameters performance and their mechanism under drought to enrich the Introduction and to explain how we chose prarmeters for simulation.
The logical consecution described as: (1) sugarcane importance, (2) problem (drought) in industry, (3) biological properties under drought, (4) application of sugarcane growth and irrigarion model, (5) importance of photosynthetic performance simulation in model, (6) chlorophyll fluorescence research revealed photosynthetic performance, (7) choice of parameter and objective of our sudy.
Point 2: The aims are too general and do not address specific novelty of the study.
Response 2: We rewrite this part to emphasize “The study provides a new perspective for sugrcane drought tolerance assessment and drought stress diagnosis” which are important for sugarcane production under seasonal drought
Point 3: Parameters PhibL, PhibH etc. - these parameter should be clearly defined at its first mention.
Response 3: Accepted experts’ opinion and carefully revised. Considering there are too many parameters in the manuscript, I have filled all the parameters used in this study into a table containing the name, definition, unit and value to make them clearly understand.
Point 4: line 131 'The Fv/Fm value and empirical coefficients...' these parameters should be described separately for better understanding.
Response 4: I have already divided the discription into two parts to make them clearly.
Point 5: 'Parameter Epsilon...' this has to be defined.
Response 5: I have filled all the parameters used in this study into a table containing the name, definition, unit and value to make them clearly understand.
Point 6: Furthermore, the fitted NPQ curve for ‘ROC16’ exhibited a flattening trend with the increasing PAR, demonstrating that the photo-protective mechanism did not involve excessive energy consumption.' This sentence is unclear.
Response 6: The discription had been revised as: we discribed the NPQ vairations for both cultivars in different day at first, then followed the concluding sentence “The results demonstrated that the photo-protective mechanism based on heat loss of ‘ROC16’ did not involve excessive energy consumption compared with ‘ROC22’”.
Point 7: A The PhiNO is not defined. To interpret these results, the concept of regulated and non-regulated non-photochemical quenching must be introduced.
Response 7: I have add these concepts in Introduction section to clarify their definition and explain the reason why we chose these parameters.
“Another two parameters, NPQ and ΦNO, were chosen in this study to analyze the energy distribution under drought. The former represented total energy dissipation into heat loss containing energy-dependent, zeaxanthin-dependent and photoinhibitory quenching [40], and the latter had been used to evaluate the occurrence of physiological damage accumulation during drought [34] and post-drought recovery [40].”
Point 8: why is a separate section 2.4 necessary? The figures should be placed where the relevant results are mentioned in the Results section.
Response 8: For the figures and tables, we accepted the experts’ opinions to move them to the right place.
Point 9: why is Fv/Fm given in 4 decimals? This is uncommon and misleading.
Response 9: Because the value of Fv/Fm we used in this study is the average value of of 9 leaves measured from different +1 leaf. I have re-rounded this value to make it reasonable.
Point 10: Model description. It is unclear why is this model selected and what was the rationale and background of its application. Therefore more information about the model and the reason of its application should be added.
Response 10: The transforming equations describing the relationship between photosynthetic quantum efficiency (Φ) and photo-thermal background were based on my dissertation which has been cited in the revised manuscript. The relationship between estimated parameters and water status was newly proposed in this study.
Point 11: line 234:Minimum, optimal, and maximum temperatures obtained from the curve fitting... how was this done?
Response 11: Indeed there is dispute of basic values of sugarcane photo-thermal limination, we intended to use our data at first, after reading the expert’s queries, we chose to summarize the basic three point temperature of sugarcane from several studies which determined as 15℃(minimum), 30℃(optimum), 40℃(maximum). Furthermore, curve fitted parameters ΦTo, ΦbL, and ΦbH, which represents the optimum Φ under To, the basic Φ under low T and high T at the limited PAR based on the new three point temperature and recaculated all the data.
Point 12: line 238-240: it is unclear what was the rationale and background of dividing quantum yield to 'basic' and 'variable' quantum yield. Furthermore, it is unclear why would basic quantum yield represent alternative electron flow. Alternative electron flow activity should be determined with simultaneous measurements of Chl fluorescence and oxygen uptake. It should be explained how the alternative electron flow pathways were determined using other combined techniques e.g. oxygen polarography and Photosystem I kinetics, because these pathways require the application of these combined methods and integrated approaches. Therefore the interpretation of section 243-248 should be reconsidered and rewritten accordingly.
Response 12: I did not discript this part clearly, the model here is not for study of electron flow pathways. Here we introduced basic quantum yield only for equation curvefitting due to the fact that Φ remained over 50% when there is no obversved Pn in previous [Flexas, 2002] and our study.
We transform the equation by using ΦTo and Φb which is clearly and can eliminate the misunderstanding. Then we explatin this phenomenon at Discussion section: “Unlike Pn,max, the occurrence of Φb (ΦbL or ΦbH) at low or high temperatures was associated with alternative electron fluxes beyond photochemical reaction processes, such as photorespiration, Mehler reaction, cyclic and pseudocyclic electron transport under normal or drought conditions [44-45]. For instance, photorespiration and Mehler reaction constituted 20% and 30% of the total electron flux, respectively [46-47]. Moreover, rETR remained at 50% under normal conditions even when photosynthesis ceased due to stomatal closure under drought stress [48]. This could also explain the existence of Φb and the gradual decrease of Φ with the increase or decrease of temperature in our study compared with Pn,max-Ta of the C3, C4, and CAM plants [49]. ”.
Point 13: line 305-306: why these cultivars? What is the novelty and significance of plant material selection?
Response 13: We chose these two tested cultivars considering the following points: (1) they were introduced and applied in production in south China as main cultivars (Wei, 2021); (2) they were widely used in many drought response researches (Yao, 2013; An,2015; Khalil, 2018). We add this information to Material and Method setion.
Point 14: line 314-315: it is unclear how were the drought experiments done.
Response 14: The drought treatment was after photothermal experiment via the natural drought stress, which is continuously withhold water and measured the data at different days after drought treatment.
Point 15: line 319: photo-thermal response model - this should be defined and described in details.
Response 15: The function of photo-thermal response model is to evaluate the Φ under different light and temperature background. We have added this information in manuscript.
Point 16: line 326: drought response model - more details are needed about this model.
Response 16: We delete this discription instead of estimating Ɛ D through the curve fitting of Φ, VWC, Fv/Fm, PAR and T with the finally aim of discribing the drought stress degree via the Ɛ D domain.
Point 17: line 339-347: It is somewhat unclear what was the rationale of the authors for using the selected chl fluorescence parameters. This could be clearer if there was a proper introduction about photosynthesis and drought stress.
Response 17: We add this information in Introduction section to explain why we chose these parameters.
“Among the chlorophyll fluorescence parameters, the reliability of Fv/Fm in evaluating the performance of sugarcane cultivars to drought stress had been verified in previous study [33]. The decreasing rate of Fv/Fm was significantly higher in drought-tolerance cultivars than that in drought-susceptible cultivars under drought stress [35]. Though Fv/Fm could become a stable drought resistance evaluation indicator for sugarcane cultivars, considering its slow and slight decrease during drought stress (mostly 2%~15%) [3,33,35], we chose to simulate another important parameter Φ (namely ΦPSII) under various water status and evaluated the sugarcane drought tolerance through the analysis of the fitted parameters resulted from two reasons: (1) Φ provides the information on the photosynthetic photon flux density and photo-inhibition based on the inactivation of PSII reaction centers [36], and exhibits a large varied amplitude (47.5%~64.3% decrease) with the aggravation of drought in C4 plants including reed and sugarcane [34,37]; (2) Φ directly related to the ETR which provides energy to biochemical reactions of CO2 assimilation [38], thus have been used to estimate photosynthetic rate via the modified FvCB model [39]. Another two parameters, NPQ and ΦNO, were chosen in this study to analyze the energy distribution under drought. The former represented total energy dissipation into heat loss containing energy-dependent, zeaxanthin-dependent and photoinhibitory quenching [40], and the latter had been used to evaluate the occurrence of physiological damage accumulation during drought [34] and post-drought recovery [40].”

Reviewer 3 Report
The manuscript reported the differential drought response strategies of drought-resistant (‘ROC22’) and –susceptible (‘ROC16’) sugarcane cultivars via photosynthetic quantum efficiency(Φ) simulation and analysis of the photosystem energy distribution, which is interesting for drought tolerance assessment or drought stress diagnosis for sugarcane cultivars.
However, this paper has the following problems, especially clarify the innovation points:
1. The introduction mainly introduces water-saving technology, which is not closely related to the research theme of this question, "Sugar cane response strategy to drought stress";
2. The structure of the article needs to be adjusted. The section "2.4 Figures, Tables and Schemes" does not need a special title. The detailed test method Table 2 should be placed in "4 Materials and Methods ";
3. The description of some charts in the text is not clear enough, such as Table 1, which makes it difficult to understand the specific meaning of the data. The description of some charts, such as Figures 3a and 3e, is not accurate enough.
Author Response
Dear editor,
Thanks for the editor’ and reviewers’ opinions, which are very helpful to improve the quality of the manuscript and the growth of young scientific personnel. We have carefully restructed the article, add explaination of the cultivars choice and experiment conduction, and a description table for all the parameters used in this study. Words in blue a the changes I have revised in the manuscript. Now I response the reviewers’ comments with a point by point and high light the changes in revised manuscript. We sincerely hope that you find our responses and modificatins satisifactory are acceptable for publication. Full details are listed below:
Point 1: The introduction mainly introduces water-saving technology, which is not closely related to the research theme of this question, "Sugar cane response strategy to drought stress"
Response 1: We accepted expert’ opinion to delet the agronomic and engenieering water saving discription instead of adding references of the chorophyll fluorescence parameters performance and their mechanism under drought to enrich the Introduction and to explain how we chose prarmeters for simulation, which is important to analyze the differental drought response in two cultivars.
The logical consecution described as: (1) sugarcane importance, (2) problem (drought) in industry, (3) biological properties under drought, (4) application of sugarcane growth and irrigarion model, (5) importance of photosynthetic performance simulation in model, (6) chlorophyll fluorescence research revealed photosynthetic performance, (7) parameter selection and objective of our sudy.
Point 2: The structure of the article needs to be adjusted. The section "2.4 Figures, Tables and Schemes" does not need a special title. The detailed test method Table 2 should be placed in "4 Materials and Methods "
Response 2: We have already adjusted the structure of the entire article. I put the discription of model construction in the Materal and Method section, rewrite the Discussion section, add references to discuss the variation of ChlF parameters under drought comparing with other researches, and move the cross analysis to Result 2.4. Please review the rest details in the revised manuscript.
For the figures and tables, we accepted the experts’ opinions to move them to the right place.
Point 3: The description of some charts in the text is not clear enough, such as Table 1, which makes it difficult to understand the specific meaning of the data. The description of some charts, such as Figures 3a and 3e, is not accurate enough.
Response 3: We reviesed the unclear description carefully and add the information for them. Take Figrues 3 as an example, “Figure 3. Excitation light energy distribution in the two tested cultivars at 1, 3, 5, 7 and 9 day after drought treatment. (Figure 3b-e share the same legend as figure 3a )”

Round 2
Reviewer 1 Report
Dear authors,
thank you for elaborating and clarifying some comments. A large part of the comments have been incorporated, but I am still sending a few minor comments for incorporation.
Point 5: In my opinion, there is a difference between comparing certain properties (as in most of the references in the reply) and building a model. The model is always relatively general and needs to be validated precisely to determine how widely it can actually be used. It can of course be published without major validation, but the discussion or conclusion section should indicate directions for further testing. I suggest you put this in a sentence in one of the chapters.
Point 10: Still not clear what „High temp”, and „Low temp” mean. Daily maximum, minimum? (If yes, please, use these more acceptable terms!). If the temperature and PAR values are average of all experiment, the daily values should be shown as a supplementary material (in a graph or table). Especially for the PAR value, the maximum may have been for a moment, so this data may not be of great significance, especially if you also look at the variance from place to place. (There is not noted, if there are average or mean values, and near to that ED or SE values...?) In the table, I don't think you need to write "during experiment" everywhere, just write it in the comment.
ROW 743: What does it mean: „+1 leaf”? (Please, write in more detail in the text, like this it is not clear, what did you want to tell)
Row 619-621: Please check the grammatical correctness of the sentence („Because….”).
Author Response
Dear editor,
Thanks for the editor’ and reviewers’ opinions, which are very helpful to improve the quality of the manuscript and the growth of young scientific personnel. We have carefully restructed the article, add explaination of the cultivars choice and experiment conduction, and a description table for all the parameters used in this study. Words in blue a the changes I have revised in the manuscript. Now I response the reviewers’ comments with a point by point and high light the changes in revised manuscript. We sincerely hope that you find our responses and modificatins satisifactory are acceptable for publication. Full details are listed below:
Point 1: In my opinion, there is a difference between comparing certain properties (as in most of the references in the reply) and building a model. The model is always relatively general and needs to be validated precisely to determine how widely it can actually be used. It can of course be published without major validation, but the discussion or conclusion section should indicate directions for further testing. I suggest you put this in a sentence in one of the chapters.
Response 1: The experts’ opinions are very rigorous and reasonable, we accepted the suggestion and add this conception at the end of the discussion. “The accuracy, persuasiveness and the application scope of the model could be improved after more cultivars (or vaireties) participating in parameter fitting and model validation.” Indeed relative experiments were conducted on three major cultivars and several new varieties with this model applied as one aspect in drought tolerance evaluation. We expect to reveal physiological mechanism from new prospects via the mutigroup analysis combined with stomatal regulation and energy metabolism in two photosystems,
Point 2: Still not clear what „High temp”, and „Low temp” mean. Daily maximum, minimum? (If yes, please, use these more acceptable terms!). If the temperature and PAR values are average of all experiment, the daily values should be shown as a supplementary material (in a graph or table). Especially for the PAR value, the maximum may have been for a moment, so this data may not be of great significance, especially if you also look at the variance from place to place. (There is not noted, if there are average or mean values, and near to that ED or SE values...?) In the table, I don't think you need to write "during experiment" everywhere, just write it in the comment.
Response 2: Indeed it is hard to find a proper method to describe this part. It occupies too much space to describe them by using either figure formate or table formate. I describe the environmental background during experiments as: everage daily high temperature (low temperature, PARmax, PARave). Please help us consider whether this statement is appropriate.
The orginal data shows as below:
|
Exp1 |
date |
High temperature |
Low temperature |
PARmax |
PAReverage |
|
photo-thermal |
2020/4/9 |
21.95 |
17.94 |
781.6 |
81.6 |
|
2020/4/10 |
25.2 |
19.45 |
1985 |
255.5 |
|
|
2020/4/11 |
25.6 |
20.82 |
1665 |
332.7 |
|
|
|
average |
24.25 |
19.40333 |
1477.2 |
223.2666667 |
|
|
stdev |
2.001874 |
1.440567 |
623.2933 |
128.6158751 |
|
|
2020/4/12 |
27.78 |
21.74 |
1843 |
291.9 |
|
drought |
2020/4/13 |
24.88 |
15.94 |
2131 |
497.1 |
|
2020/4/14 |
22.67 |
13.66 |
2106 |
291.8 |
|
|
2020/4/15 |
23.1 |
14.15 |
1203 |
203.8 |
|
|
2020/4/16 |
25.66 |
15.3 |
2305 |
461 |
|
|
2020/4/17 |
27.58 |
21.02 |
2181 |
495.4 |
|
|
2020/4/18 |
27.08 |
22.78 |
1888 |
360.2 |
|
|
2020/4/19 |
26.61 |
23.42 |
1852 |
168.3 |
|
|
2020/4/20 |
29.85 |
24.29 |
2025 |
391.3 |
|
|
2020/4/21 |
29.92 |
24.92 |
2064 |
344.8 |
|
|
2020/4/22 |
28.52 |
24.52 |
1201 |
193.7 |
|
|
|
average |
26.587 |
20 |
1895.6 |
340.74 |
|
|
stdev |
2.538534 |
4.672589 |
388.4397 |
123.956141 |
|
Exp2 |
….. |
….. |
….. |
….. |
….. |
|
Exp3 |
….. |
….. |
….. |
….. |
….. |
|
Exp4 |
….. |
….. |
….. |
….. |
….. |
|
Exp5 |
….. |
….. |
….. |
….. |
….. |
Point 3: What does it mean: „+1 leaf”? (Please, write in more detail in the text, like this it is not clear, what did you want to tell)
Response 3: I have described +1 leaf clearly in the followed bracket, which is “the first fully expanded leaf counting from the top down”, this leaf position had been used for the physiological indicators measurement and sampling in previous studies (Jin 2012; Zhang, 2014) and could be proved from our previous research on three major sugarcane cultivars (An, 2015).
An,D; Wei, C; Cao, J; Dou, M. The study of chlorophyll fluorescence characteristics on different leaf position of three sugarcane seedling cultivars. Chinese Journal of Tropical Crops 2015, 36(11): 2019-2027.
Point 4: Please check the grammatical correctness of the sentence („Because….”)
Response 4: I rewirte this sentence to make it correct: “This phenomenon resulted from the fact that both net photosynthetic rate and ΦPSII significantly decreased under drought-cold stress than under drought stress only [52].”

Reviewer 2 Report
The manuscript has improved, however, further revisions are necessary.
It is helpful that the authors collected all parameters in a table, however it is incomplete and contains several errors: The definition of PhiNPQ is missing. Several typos exists: 'Nonphotochemical quanching' is with error, PAR: 'photo active radiation' is incomplete term, etc. More typo errors may exists, so the authors need a comprehensive revision in grammar. The term of PhiNPQ 'down regulated energy dissipation' in line 145 is also not entirely correct, please amend it according to relevant nomenclature (e.g. Kramer et al. Photosynthesis Research 79: 209–218, 2004., Hendrickson et al. Photosynthesis Research 82: 73–81, 2004.)
line 164-165: Φ provides the information on the photosynthetic photon flux density - how would quantum efficiency of PSII provide information about the photon flux density? This has to be supported with relevant references.
Define in brief the FvCB model.
Figure 1 legend: ΦbH is not defined.
Figure 3 (Figure 3b-e share the same legend as figure 3a ): if this is the case, why did the trend completely reverse on day 9? The quantum yield increases and NPQ decreases with PAR, it seems the symbols are swapped or mislabeled compared to the other panels.
line 379-381> it is unclear whether these changes are significant, because epsilon D and quantum yield appears to be the same in ROC22 and ROC16. NPQ is indeed higher in ROC22 at the onset of drought.
line 383-385> However, ‘ROC16’ maintained a relatively higher Ɛ D to ensure photochemical efficiency until the rSWC dropped to 29%, after which the NPQ increased merely at PAR 100~900 μmol·m-2·s-1. I am not sure which data supports this. Epsilon D seems to be the opposite, higher in ROC22 at 29% water content.
line 639> term 'productive energy loss' is unclear.
line 742 - 757: it is unclear how was the chlorophyll fluorescence parameters obtained as a function of PAR. Was it with the light curve protocol? Then it also should be described how many light steps were used, with what PAR levels and how long were the individual light steps.
It is still somewhat unclear that the presented results (i.e. Fig1-3) represent which experiment(s). Five experiments are mentioned, but it is unclear which of these experiments are displayed on the figures or whether the presented results are the average of experiments 1-3 (the photo-thermal model). This should be clarified.
The model description and validation appears to be based on a PhD thesis, however I could not find this information. It is strongly recommended to add this information as supplementary material or appendix or data repository, otherwise the model cannot be evaluated and applied.
Author Response
Dear editor,
Thanks for the editor’ and reviewers’ opinions, which are very helpful to improve the quality of the manuscript and the growth of young scientific personnel. We have carefully restructed the article, add explaination of the cultivars choice and experiment conduction, and a description table for all the parameters used in this study. Words in blue a the changes I have revised in the manuscript. Now I response the reviewers’ comments with a point by point and high light the changes in revised manuscript. We sincerely hope that you find our responses and modificatins satisifactory are acceptable for publication. Full details are listed below:
Point 1: It is helpful that the authors collected all parameters in a table, however it is incomplete and contains several errors: The definition of PhiNPQ is missing. Several typos exists: 'Nonphotochemical quanching' is with error, PAR: 'photo active radiation' is incomplete term, etc. More typo errors may exists, so the authors need a comprehensive revision in grammar. The term of PhiNPQ 'down regulated energy dissipation' in line 145 is also not entirely correct, please amend it according to relevant nomenclature (e.g. Kramer et al. Photosynthesis Research 79: 209–218, 2004., Hendrickson et al. Photosynthesis Research 82: 73–81, 2004.)
Response 1: Thanks to experts’ for carefully pointing out the mistakes. We revised all the description of the parameters according to the previous published researches. The definition of ΦNPQ is in the Introduction section.
Point 2: Φ provides the information on the photosynthetic photon flux density - how would quantum efficiency of PSII provide information about the photon flux density? This has to be supported with relevant references.
Response 2: This statement is inaccurate, I rewrite it as “Φ provides the information on noncyclic electron transport rate through PSII” according to Baker’s (2008) paper.
Point 3: Define in brief the FvCB model.
Response 3: I have already supplimented this information by cited another reference produced by the same author, which used a biochemical C4 photosynthesis model and combined gas exchange and chlorophyll fluorescence measurements to estimate bundle-sheath conductance.
Point 4: Figure 1 legend: ΦbH is not defined
Response 4: I add this information on the chart description.
Point 5: (Figure 3b-e share the same legend as figure 3a ): if this is the case, why did the trend completely reverse on day 9? The quantum yield increases and NPQ decreases with PAR, it seems the symbols are swapped or mislabeled compared to the other panels.
Response 5: I made a mistake of the legend used in chart making. I have replaced this figure with revised version.
Point 6: line 379-381> it is unclear whether these changes are significant, because epsilon D and quantum yield appears to be the same in ROC22 and ROC16. NPQ is indeed higher in ROC22 at the onset of drought.
Response 6: I separated two figures to make this statement clearly, figure 2b was used for higher NPQ. From figure 3c we could infer that the ƐD of ROC16 was slightly lower than that of ROC22 when their rSWC dropped to 40% level.
It is hard to find the specific light and temperature background for each measuement at field condition. We did not have the data measured at the same given PAR and temperature under the same water status with variance analysis, instead of that, and considering about the future application at field condition, we adopted the method of curve fitting for multi-point measurement, then compared the fitting curve and parameters for different cultivars.
From another point of view, the fitted curve of rETR caculated from the Φ measured at day 3 showed that ROC16 was higher than ROC22. This could also prove the electron transprot capacity of ROC16 was higher than that of ROC22, indicated ROC16 had relatively lower ƐD compared with ROC22. (not shown in this manuscript)
Point 7: line 383-385> However, ‘ROC16’ maintained a relatively higher Ɛ D to ensure photochemical efficiency until the rSWC dropped to 29%, after which the NPQ increased merely at PAR 100~900 μmol·m-2·s-1. I am not sure which data supports this. Epsilon D seems to be the opposite, higher in ROC22 at 29% water content.
Response 7: This is a mistake in writing and should be revised as “lower”, bucause low Ɛ value represented a small change of Φ with increasing PAR indicating higher light efficiency.
Point 8: line 639> term 'productive energy loss' is unclear.
Response 8: I revised this sentence as “……, which probably represented the heat energy loss in protection of the stressed younger leaves [65].” According to the reference, both productive and protective dissipation belongs to heat energy, which could be indicated by an increase in ΦNPQ and decline in ΦNO occurred simultaneously.
Point 9: line 742 - 757: it is unclear how was the chlorophyll fluorescence parameters obtained as a function of PAR. Was it with the light curve protocol? Then it also should be described how many light steps were used, with what PAR levels and how long were the individual light steps.
Response 9: I add the measurement’s details to make it clearly. The chlorophyll fluorescence parameters measured under natural light conditions except for drought treatment measured in phytotron in Exp5. During the measurement, we adjust the clip to make sure a relatively constant PAR (±10) and waited for a stable Fs to start the light-adapted measurement. After the dark adapted parameters measurement, the leaves was put in light envrionment for 30 min before the next round of Fs and Fm’ measurement.
In phytotron, we had to adjust the distance between the measured leaves and light source resulted from the light in phytotron exhibited a large decay with the increase of the distanse between the canopy and the light source. We try our best to make the measuring points evenly distributed from 0 PAR to 1200 μmolm-2s-1 PAR.
Point 10: It is still somewhat unclear that the presented results (i.e. Fig1-3) represent which experiment(s). Five experiments are mentioned, but it is unclear which of these experiments are displayed on the figures or whether the presented results are the average of experiments 1-3 (the photo-thermal model). This should be clarified.
Response 10: This information had been stated in Material and Method section. The data of Figure 1 came from the photo-thermal Exp1-3. The data of figure 2 & 3 came from the drought Exp5. The data of figure 4a and 4b came from photo-thermal Exp4-5 and drought Exp1-4, respectively.
I have considered about to add this information behind each chart description, but I wonder if they are appropriate?
Point 11: The model description and validation appears to be based on a PhD thesis, however I could not find this information. It is strongly recommended to add this information as supplementary material or appendix or data repository, otherwise the model cannot be evaluated and applied.
Response 11: Only calibration equation of environmental photo-thermal background on Φ had been cited in this study. I have already put my dissertation in reference. “An, D.S. Chlorophyll fluorescence based water stress diagnosis for cut Lilium grown in greenhouse [D]. Nanjing Agriculture University 2011” . I have its electronic version, but it was written in Chinese only with English title and abstract.

Reviewer 3 Report
I suggest to change the title a little, such as removing the "(drought-resistant and –susceptible)" and adding some words of measuring chlorophyll fluorescence parameters.
Author Response
Dear editor,
Thanks for the editor’ and reviewers’ opinions, which are very helpful to improve the quality of the manuscript and the growth of young scientific personnel. We have carefully restructed the article, add explaination of the cultivars choice and experiment conduction, and a description table for all the parameters used in this study. Words in blue a the changes I have revised in the manuscript. Now I response the reviewers’ comments with a point by point and high light the changes in revised manuscript. We sincerely hope that you find our responses and modificatins satisifactory are acceptable for publication. Full details are listed below:
Point 1: I suggest to change the title a little, such as removing the "(drought-resistant and –susceptible)" and adding some words of measuring chlorophyll fluorescence parameters.
Response 1: As for the (drought-resistant and –susceptible), I think keeping this statement might be more beneficial to clarify our purpose. Both photosynthetic quantum efficiency and energy distribution were based on chlorophyll fluorescence parameters analysis.
